# Tomato Oil Encapsulation by α-, β-, and γ-Cyclodextrins: A Comparative Study on the Formation of Supramolecular Structures, Antioxidant Activity, and Carotenoid Stability

**DOI:** 10.3390/foods9111553

**Published:** 2020-10-27

**Authors:** Miriana Durante, Francesco Milano, Monica De Caroli, Livia Giotta, Gabriella Piro, Giovanni Mita, Mariaenrica Frigione, Marcello Salvatore Lenucci

**Affiliations:** 1Institute of Sciences of Food Production (ISPA), CNR, Monteroni di Lecce, 73100 Lecce, Italy; miriana.durante@ispa.cnr.it (M.D.); francesco.milano@ispa.cnr.it (F.M.); giovanni.mita@ispa.cnr.it (G.M.); 2Department of Biological and Environmental Sciences and Technologies (DiSTeBA), University of Salento, Monteroni di Lecce, 73100 Lecce, Italy; monica.decaroli@unisalento.it (M.D.C.); livia.giotta@unisalento.it (L.G.); gabriella.piro@unisalento.it (G.P.); 3Department of Innovation Engineering, University of Salento, Via Arnesano, 73100 Lecce, Italy; mariaenrica.frigione@unisalento.it

**Keywords:** bioactive compounds, cyclodextrinosomes, green-chemistry, lycopene, Pickering emulsions, plant antioxidants, shelf-life, spectroscopic characterization, supercritical fluid extraction, Trolox equivalent antioxidant capacity (TEAC) assay

## Abstract

Cyclodextrins (CDs) are oligosaccharides, comprising 6 (α), 7 (β), or 8 (γ) glucose residues, used to prepare oil-in-water emulsions and improve oil stability towards degradation. In this research, the aptitude of α-, β-, and γ-CDs to form complexes with a supercritical CO_2_ extracted lycopene-rich tomato oil (TO) was comparatively assessed. TO/CD emulsions and the resulting freeze-dried powders were characterized by microscopy, Fourier transform infrared-attenuated total reflection (FTIR-ATR), and differential scanning calorimetry (DSC), as well as for their antioxidant activity. Furthermore, carotenoid stability was monitored for 90 days at 25 and 4 °C. Confocal and SEM microscopy revealed morphological differences among samples. α- and β-CDs spontaneously associated into microcrystals assembling in thin spherical shells (cyclodextrinosomes, Ø ≈ 27 µm) at the oil/water interface. Much smaller (Ø ≈ 9 µm) aggregates were occasionally observed with γ-CDs, but most TO droplets appeared “naked”. FTIR and DSC spectra indicated that most CDs did not participate in TO complex formation, nevertheless structurally different interfacial complexes were formed. The trolox equivalent antioxidant capacity (TEAC) activity of emulsions and powders highlighted better performances of α- and β-CDs as hydrophobic antioxidants-dispersing agents across aqueous media. Regardless of CDs type, low temperature slowed down carotenoid degradation in all samples, except *all*-[*E*]-lycopene, which does not appear efficiently protected by any CD type in the long storage period.

## 1. Introduction

Among the numerous bioactive compounds that make tomato (*Solanum lycopersicum* L.) a functional food, carotenoids are the group to which the health-protective properties are distinctively attributed. Lycopene (*ψ*,*ψ*-carotene) is the most abundant carotenoid found in the red varieties, contributing to 90–96% of total pigments at full ripening, predominantly in the *all*-[*E*] isomeric conformation [1].

As a powerful lipophilic natural antioxidant and anti-inflammatory molecule, lycopene dietary intake is associated with a decreased prevalence of several chronic pathologies, including cardiovascular diseases, arthritis, asthma, neurodegenerative disorders, type 2 diabetes, and various cancers [2]. A lycopene consumption between 5 and 7 mg day^−1^ has been suggested to have protective effects on healthy adults [3]. However, in disordered conditions, such as overt prostate neoplasia, daily doses of up to 75 mg seem required to reduce further cancer progression, though the evidence of clinical trials is still not conclusive [4].

Due to the growing interest and awareness on the health benefits of carotenoids from tomatoes by food, personal care and cosmetic, nutraceutical and pharmaceutical companies, lycopene extraction market is expected to expand rapidly [5]. Most commercial lycopene is extracted from tomato with conventional methods using organic solvents approved for food applications (e.g., ethanol or ethyl acetate). Nevertheless, the rising concerns for potential toxicity of such solvents, even trace amounts, is pushing for the application of green production strategies. Supercritical-CO_2_ (SC-CO_2_) technology demonstrated high efficiency for carotenoid extraction from a range of plant materials and/or by-products, including entire tomato (>90% yield), with no need for organic solvents [6,7,8]. The resulting oil contains high concentrations of carotenoids, tocochromanols, phytosterols, and polyunsaturated fatty acids and has broad marketability as high-quality food grade pigment and additive in the formulation of pharmaceuticals, cosmeceuticals, nutraceuticals, and functional foods [9,10,11,12,13].

Among carotenoids, the characteristic acyclic, highly conjugated polyene structure (with 11 out of 13 conjugated double bonds) makes lycopene particularly susceptible to oxidative degradation especially when exposed to light, oxygen, high temperatures, extreme pH, and surface-active molecules [14]. These factors prompt *all*-[*E*]-lycopene isomerization to any of a number of [*Z*]-isomers with even reduced relative stability (5-[*Z*] ≥ *all*-[*E*] ≥ 9-[*Z*] ≥ 13-[*Z*] > 15-[*Z*] > 7-[*Z*] > 11-[*Z*]) [15]. These aspects, together with the extremely high lipophilicity (clog *p* = 17.64) and the resulting very low aqueous solubility of lycopene, are significant barriers to its oral formulation and bioavailability [2,16]. In this respect, incorporation in liposomes, molecular inclusion, nano- and micro-encapsulation techniques have been used to reduce lycopene susceptibility against isomerization/degradation, simultaneously improving water solubility, absorption, and bioavailability [17,18,19,20].

Cyclodextrins (CDs) are a family of biocompatible and nontoxic starch derived α-(1,4)-oligosaccharides, comprising 6 (α), 7 (β), or 8 (γ) glucose residues with well-established use as encapsulating agents. CDs are tasteless, odorless, and noncaloric molecules, exert very low hygroscopicity and high thermal stability (up to 100 °C), thus find wide industrial application to increase solubility, stability, and bioavailability of hydrophobic bioactive molecules in food, nutraceutical, cosmeceutical, and pharmaceutical formulations [21,22,23,24]. Furthermore, CDs are widely used as food additives for flavor stabilization, elimination of unpleasant tastes or undesired compounds, and to avoid microbiological contamination and browning reaction [25].

CDs have a torus-shaped structure with a hydrophilic external surface stiffened by hydrogen-bonding between the 3-OH and 2-OH groups of the glucose residues and a relatively hydrophobic internal cavity lined by hydrogen atoms and glycosidic oxygen bridges carrying the nonbonding electron pairs directed towards the inside [26]. CDs are prepared from starch by enzymatic hydrolysis using a bacterial amylase (cyclodextrin glycosyl transferase) and purified by crystallization from the aqueous solution. Thus, commercial CDs represent undefined hydrates where the entrapped water contributes to stabilize the crystal lattice [27].

CDs can interact with single molecules or organic mixtures forming different types of complexes [28]. Simple inclusion complexes spontaneously assemble, either in the solid or in solution, when a lipophilic guest molecule enters the internal cavity of CDs. In this process, the enthalpy rich water molecules within the CDs’ internal cavity are completely or partially displaced, while the inclusion complex is rendered thermodynamically stable via hydrophobic forces, van der Waals interactions or hydrogen bonds [29,30,31]. Noninclusion complexes by hydrogen bonding between certain guest molecules and the external rim of CDs are also formed, as well as supramolecular aggregates capable of dissolving drugs through micelle-like structures [32,33].

The use of CDs to prepare surfactant-free oil in water emulsions and improve stability of oil constituents towards thermal-, photo-, and oxidative-degradation has been explored specifically aimed at overcoming the negative effects of some detergents on humans or the environment (e.g., skin irritation, hemolytic properties, poor biodegradability, etc.). In aqueous media, the presence of dispersed oil triggers the formation of inclusion complexes at the oil/water interface, which spontaneously associate into micro- and nanocrystals of different diameters (typically between 20 and 200 nm) and shapes, depending upon the relative concentration of CDs and oil in the aqueous phase, as well as the oil composition [34,35]. The adsorption of such solid particles at the oil/water interface, forming spherical (cyclodextrinosomes) or nonspherical shells around the oil droplets, stabilizes the resulting Pickering emulsions [34,36,37,38].

In a previous work, we highlighted differential effects of α-CDs coating on the storage stability of a lycopene-rich tomato oil extracted by SC-CO_2_ (thereafter indicated simply as tomato oil (TO)). In particular, though isoprenoid retention was generally improved, lycopene resulted more susceptible to oxidation than in crude TO [9]. Hence, in this research, the aptitude of α-, β-, and γ-CDs to form Pickering oil/water emulsions with TO and delay carotenoids degradation was comparatively assessed for the first time. The CD-stabilized TO/water emulsions (from now on called for simplicity TO/CD emulsions) and/or the corresponding dehydrated material (powders) obtained upon freeze-drying were characterized by microscopy, spectroscopy, and calorimetry. Their antioxidant activity was also measured by the Trolox equivalent antioxidant capacity (TEAC) assay and compared with that of crude TO. Lastly, carotenoid stability was monitored over 90 days at 4 and 25 °C, in the dark.

## 2. Materials and Methods

### 2.1. Chemicals

Carotenoid standards were purchased from CaroteNature (Lupsingen, Switzerland). CAVAMAX^®^ W6, W7 and W8 (α, β and γ, respectively) food grade CDs were kindly provided by IMCD Italy SpA (Milano, Italy). High purity carbon dioxide (CO_2_—99.995%) for supercritical fluid extraction was purchased from Mocavero Ossigeno (Lecce, Italy), high performance liquid chromatography (HPLC) grade solvents were bought from Sigma-Aldrich (Milan, Italy).

### 2.2. Tomato Oil Encapsulation Process

A tomato matrix was prepared as previously reported by Lenucci et al. [9] from red-ripe tomato berries of the high lycopene cultivar Heinz 1113 grown in open field in the province of Foggia (southern Italy). Briefly, tomatoes were blanched in water at 70 °C for 5 min, crushed and sieved by a Reber 9004 N tomato squeezer (Reber, Luzzara, Italy) to obtain a purée. The purée was centrifuged at 27,000× *g* for 10 min. The pellet was dehydrated to constant weight by a Christ ALPHA 2–4 LSC freeze-dryer (Martin Christ Gefriertrocknungsanlagen GmbH, Osterode am Harz, Germany). Simultaneously, the tomato seeds were recovered and vacuum dried at 60 °C by a Salvis Lab IC40 vacuum-drying oven (Bio Instruments S.r.l., Firenze, Italy). Freeze-dried tomato pellet and dehydrated seeds were ground in a laboratory ultra-centrifugal mill (ZM200, Retsch GmbH, Haan, Germany) through 35 mesh and 18 mesh sieves, respectively, and blended in a ratio of 1:1 by weight to obtain the dehydrated tomato matrix used for SC-CO_2_ oil extraction. Aliquots (25 g) of the dehydrated tomato matrix were packed into a 25 mL stainless-steel extraction vessel (ø = 1 cm^2^; h = 25 cm) and dynamically extracted by a Speed Supercritical Fluid Extraction (SFE) system (Applied Separations, Allentown, PA, USA) for 3 h at 50.3 MPa, 86 °C, by flowing CO_2_ at a rate of ≈ 0.5 kg/h.

α-, β-, and γ-CDs were dissolved in 20 mL distilled water to a 10 mM final concentration. All CDs were solubilized at 25 °C, except β-CDs that required heating at 40 °C for 1 h. Tomato oil (0.1 g) was slowly added to the aqueous solution under continuous stirring and nitrogen sparging. Stirring continued for 24 h under nitrogen allowing emulsions to form. All emulsions stabilized by α-, β-, and γ-CDs were prepared in triplicate and poured in clean tubes before use. Aliquots of the emulsions were freeze-dried to constant weight to obtain a solid material thereafter identified as TO/α-, β-, or γ-CD powder. In our experiments, we kept both CD concentration (10 mM) and total oil volume fraction (ɸ_0_ = 0.05) constant.

### 2.3. Carotenoid Entrapment Efficiency (EE%)

Each TO/CD powder (20 mg) was washed twice with 2 mL distilled water and centrifuged at 5000× *g* for 20 min. Carotenoids were extracted with 1 mL acetone containing 0.05% of butylated hydroxytoluene (BHT), 1 mL ethanol 95% (*w*/*w*), and 2 mL hexane. Samples were vigorously stirred (1 min, 3000 rpm) and sonicated for 1 min in a Labsonic177 LBS1-10 ultrasonic bath (Falc Instruments, Treviglio, Italy). Then, 3 mL distilled water was added. The organic phase was separated by centrifugation (5000× *g*, 10 min). The aqueous phase was extensively re-extracted with 2 mL hexane until all pigments were removed (five extractions). The organic phases were collected, dried under nitrogen, and assayed for carotenoids.

Triplicate aliquots (0.1 g) of TO and the whole extract from CDs were dissolved in 1 mL ethyl acetate, filtered through a 0.45 µm syringe filter (Millipore Corporation, Billerica, MA, USA) and analyzed by HPLC as previously reported [9]. Briefly, an Agilent 1100 Series HPLC system equipped with a reverse-phase C30 column (5 µm, 250 × 4.6 mm) (YMC Inc., Wilmington, NC, USA) was used. The mobile phases were methanol (A), 0.2% ammonium acetate aqueous solution/methanol (20/80, *v*/*v*) (B) and tert-methyl butyl ether (C). The gradient elution was: 0 min, 95% A and 5% B; 0–12 min, 80% A, 5% B and 15% C; 12–42 min, 30% A, 5% B and 65% C; 42–60 min, 30% A, 5% B and 65% C; 60–62 min, 95% A, and 5% B. Absorbance was registered by diode array at wavelengths of 475 nm. Carotenoids were identified by comparing their retention times and ultraviolet–visible (UV–Vis) spectra to those of authentic isoprenoid standards and quantified based on their calibration curves.

The EE% of the main identified carotenoids was calculated as:EE% = (CE/CT) × 100
where CE is the amount of carotenoid extracted per unit mass of TO/CD powder and CT is the theoretic amount of carotenoids per unit mass of TO/CD powder calculated based on the carotenoid content of crude TO added to the CDs.

### 2.4. Characterization of TO/α-, β-, and γ-CD Emulsions and Powders

#### 2.4.1. Fourier Transform Infrared-Attenuated Total Reflection (FTIR-ATR) and Differential Scanning Calorimetry (DSC) Analyses

Fourier transform infrared (FTIR) spectra were recorded using a Spectrum One spectrometer (Perkin-Elmer, Waltham, MA, USA) equipped with a deuterated triglycine sulphate (DTGS) detector. Spectra were collected operating in attenuated total reflection (ATR) mode, with a resolution of 4 cm^−1^ using as internal reflection element (IRE) a three-bounce 4-mm diameter diamond microprism (SensIR Technologies, Danbury, CT, USA). The native α-, β-, and γ-CDs and the resulting TO/CD powders were deposited directly onto the IRE carefully covering the upper face of the diamond crystal and using a mechanical press for ensuring a tight contact with the IRE. Moreover, FTIR spectroscopy was employed for characterizing the continuous and dispersed phases of TO/CD emulsions. For this purpose, a centrifugation step at 7000× *g* for 2 min was carried out for separating the aqueous phase from the CD crystals-encapsulated oil phase that sedimented on the bottom of the tube. TO/CD films were prepared casting 3 µL of wet pellet onto the IRE and allowing the solvent to evaporate. As for freeze-dried samples, FTIR-ATR spectra of TO/CD cast films were acquired using the bare IRE spectrum as background averaging 16 interferograms. The aqueous continuous phase of each Pickering emulsion (supernatant) was analyzed against a water background averaging 64 interferograms.

Spectra of standard CDs solutions at increasing concentration were collected in the same modality using water as background, obtaining a linear relationship between the intensity of the peak at 1030–1036 cm^−1^ and CD concentration in the range 0.2–50 mM for α- and γ-CDs and 0.2–10 mM for β-CDs. The intensity of the specific marker peak in the spectra of supernatants of centrifuged TO/CD emulsions was converted into CD concentration using the calibration curve, allowing to quantify the amount of CD involved in encapsulation of TO droplets.

The thermal behavior of free CDs, TO, and TO/CD powders was studied by differential scanning calorimetry (DSC 822 Mettler Todedo, Columbus, OH, USA) calibrated with a pure indium standard. Samples (from 5 to 10 mg), sealed in aluminum pans, were examined in a range of temperature of 25–300 °C. An inert atmosphere (nitrogen, flow rate: 80 mL/min) and a constant heating rate of 10 °C/min were employed during the dynamic thermal scan. The results of, at least, three independent experiments, were averaged.

#### 2.4.2. Scanning Electron Microscopy (SEM) and Laser Confocal Scanning Microscopy (LCSM)

For SEM imaging, single drops of TO/α-, β-, and γ-CD emulsions, or small amounts of the corresponding freeze-dried powders, were deposited on conductive carbon adhesive stubs (Ø = 9 mm). Emulsions were air-dried overnight. All samples were gold sputtered (40 nm layer thickness) under high vacuum with a Balzers SCD 040 sputter coater (BAL-TEC AG, Balzers, Lichtenstein). Microstructure observation were carried out using a Helios 600 i Dual Beam SEM/Focused ion beam (FIB) (FEI Technologies Inc., Hillsboro, OR, USA) operated under high vacuum at an accelerating voltage of 5.00 kV.

For LCSM, 100 µL of TO/α-, β-, and γ-CD emulsions were gently mixed with 10 µL of melted 1% agarose solution. The emulsion was rapidly loaded by capillarity between a coverslip and a microscope slide glass separated by 0.17 mm spacers, allowed to form a gel, and observed with a LSM 710 Zeiss (ZEN Software, GmbH, Germany) using excitation wavelengths of 488 and 543 nm. Lycopene autofluorescence was detected within the 500–550 nm wavelength range, assigning the red color.

### 2.5. Antioxidant Activity Measurements

The antioxidant activity of TO and TO/CD emulsions and powders was evaluated through the attenuation of the preformed ABTS^•+^ (2,2′-Azino-bis(3-ethylbenzothiazoline-6-sulfonic acid) diammonium salt) radical. The colored radical was prepared following the procedure described in a previous work [17] and suitably diluted to 0.7 absorbance at 734 nm. TO was dissolved in hexane at 50 mg/mL and further diluted 1:20, 1:40, and 1:80 for actual measurement. A stock solution of α-tocopherol, used as hydrophobic antioxidant standard, was also prepared in hexane. A total of 50 µL of standard at increasing concentration, and TO at different dilutions were added to 950 µL radical solution in Eppendorf tubes. After 1 h incubation, the tubes were centrifuged for 2 min at 7000× *g* and the top layered organic phase was removed. Finally, the absorbance was measured at 734 nm for all samples and the antioxidant activity of TO was given as α-Tocopherol Equivalents per mass of oil. For TO/CD emulsions and TO/CD freeze-dried powders, the same procedure was applied, except that Trolox was used as hydrophilic antioxidant standard dissolved in PBS (phosphate buffered saline). All TO/CD emulsions were analyzed without dilution. After 1 h incubation step, the solutions were centrifuged 2 min at 7000× *g* to pellet the suspended α-, β-, and γ-CD encapsulated TO droplets, the absorbance was measured at 734 nm for all samples and the antioxidant activity of relevant emulsions was given as Trolox Equivalents per mass of sample. For TO/α, β-, and γ-CD powders, samples were suspended at 10 mg/mL concentration in PBS and the TEAC was assessed following the same procedure outlined for the emulsions. Furthermore, to evaluate the time dependence of TEAC and determine possible differences among emulsions or between emulsions and powders, the antioxidant activity of some representative samples (crude TO, TO/α, β-CD powders, and TO/α-CD emulsion) was monitored up to 180 min.

### 2.6. Storage of TO/CD Emulsions and Powders

Triplicate aliquots from each TO/CD emulsion and powder (500 and 200 mg, respectively) were placed in 2 mL amber glass vials, sealed with a Polytetrafluoroethylene (PTFE)/Silicone septa silver aluminum crimp cap (the headspace above the samples consisting of the entrapped air) and stored in different conditions of temperature (4 or 25 °C) and dark presence. At predetermined time points (every 15 days up to a maximum of 120 days), from each aliquot kept in the different storage conditions, TO was eluted and assayed for carotenoids, during 180 days. Simultaneously, aliquots (100 mg) of crude TO, subjected to the same storage conditions, were monitored. Carotenoids were extracted and characterized as described above.

### 2.7. Kinetic Analysis

The data were best fit by a first-order kinetic model according to the following equation:ln(C) = ln(C_0_) − k·t
where C is the concentration (µg/g) of each molecule in crude TO, TO/α-, β-, γ-CD emulsions and relevant freeze-dried powders at storage time t; C_0_ is the concentration (µg/g) of each molecule at the initial time (before storage), t is the storage time (day), k is the degradation rate constant (day^−1^). Degradation rate constants (k) were obtained from the slope of a plot of the natural log (ln) of the percentage of carotenoids vs. time.

The half-life (t_1/2_) was determined at a specific temperature by the equation:t_1/2_ = ln(2)/k

### 2.8. Statistical Analysis

The data are presented as the mean value of three independent replicate experiments (*n* = 3) with standard deviation. One-way ANOVA (Analysis of Variance) with Tukey’s post-hoc test was applied to establish significant differences between means (*p* < 0.05). All statistical comparisons were performed using SigmaStat version 11.0 software (Systat Software Inc., Chicago, IL, USA).

## 3. Results and Discussion

### 3.1. Entrapment Efficiency of Carotenoids in the TO/CD Powders

Like all vegetable oils, TO is mostly composed of triglycerides together with di- and monoglycerides, as well as free fatty acids. It also contains several other bioactive lipophilic micronutrients (e.g., carotenoids and tocochromanols) responsible for many well-known health-promoting properties. We previously described the quali-quantitative profile of carotenoids in TO and highlighted marked differences among batches depending on tomato genotype, stage of ripening, as well as pre- and post-harvest treatments and processing [8,9,39,40]. Table 1 reports the concentrations of the main carotenoids in the batch of crude TO used in this research, the carotenoid profiles of TO/α-, β-, and γ-CD powders and the calculated EE%. *All*-[*E*]-lycopene, [*Z*]-lycopene isomers, and β-carotene (615.4, 229.2, and 34.1 mg/100 g, respectively) were the main carotenoids in the crude TO with *all*-[*E*]-lycopene contributing about 70% of the total, in agreement with our previous reports [9,39].

The EE% of total carotenoids varied over a range of 44–62%. The highest EE% of TO/β-CD powder was in agreement with Piyawan et al. [41], who demonstrated that Gac fruit (*Momordica cochinchinensis* Spreng) aril extract in β-CDs (1:1 concentration ratio) produced a higher EE% of lycopene and β-carotene than other carriers (maltodextrin or gelatin). The EE% of the individual carotenoids was higher in TO/α- and β-CDs than in TO/γ-CD powders. The results showed significant difference (*p* > 0.05) between EE% of β -carotene in TO/α- and β-CD powders (100%) and TO/γ-CD powder (68%). Lower EE% (53–62%) were reported by Gomes et al. [42] in red bell pepper (*Capsicum annuum* L.) carotenoid extracts encapsulated into CDs, whereas de Lima Petito et al. [43] observed an EE% of β-carotene over 60% using 2-hydroxypropyl-β-CD as encapsulating agent.

The EE% of [*Z*]-lycopene isomers ranged from 71% to 76% in the TO/α- and β-CD powders and was much lower (≈42%) in TO/γ-CD powder. The EE% values for *all*-[*E*]-lycopene were 44% and 57% in the TO/γ-CD and β-CD powders, respectively, lower than those reported by Wang et al. [44] in *all*-[*E*]-lycopene (98% purity)/β-CD inclusion complexes (≈72%). Lycopene purity has been reported to significantly affect EE%. A purity above 52% was required to maximize EE% to about 80% in gelatin/sucrose-based microcapsule, probably because other compounds of the oils (e.g., lipids) hinder a proper shell formation [45]. Besides, a partial degradation during emulsification and dehydration processes may also contribute to a decrease of *all*-[*E*]-lycopene EE%. In fact, for the lack of cyclic end groups at both ends and its linear structure, *all*-[*E*]-lycopene may be preferentially included in the internal cavity of CDs that form the hydrophilic shell around the oil droplet, being, at least partially, sequestered from the interaction with lipophilic stabilizing molecules and more exposed to oxygen [9].

### 3.2. Characterization of the TO/CD Emulsions and Powders

#### 3.2.1. Macroscopic Characteristics of TO/α-, β-, and γ-CD Emulsions and Powders

During the incubation of the α-, β-, and γ-CD aqueous solutions with crude TO, the oil phase tended to disperse due to the formation of TO/CD aggregates initially suspended in solution. When emulsions were left to stand, a phase separation appears in all samples (Figure 1).

In TO/α-CD emulsion a compact red sediment was readily (<0.5 h) formed leaving a clear aqueous upper phase with a floating thin oily layer, already visible immediately after stopping stirring (0 h). When β-CD was used as emulsifying agent, a larger orange sediment settled over time with no visible oily phase even after 48 h standing. Finally, a turbid creamy suspension (stable for several hours) topped by a tick deep orange oily phase and a less definite yellow sediment characterized the TO/γ-CD emulsion.

Clear differences in the color and macroscopic appearance of the samples were also noticed after freeze-drying (Figure 2). The TO/α-CD dehydrated material appeared yellow-orange, harsh with a rough dry-straw texture, TO/β-CD was, instead, an orange soft and silky powder, whereas TO/γ-CD gave a greasy and smooth compact orange-red substance strongly tingent to the touch.

The color change from yellow-orange to red-orange may arise from an increased absorption of green radiation by highly concentered carotenoids as reported by Chen et al. [46]. In our case, being the loaded amount of TO the same in CD/TO powders, the color intensity can be considered directly proportional to the “apparent” carotenoid concentration in the powder, which in turn depends on the carotenoid amount that can actually interact and absorb incident light of specific frequencies. The apparent carotenoid concentration is expected to be therefore inversely proportional to the inclusion efficiency and to the efficiency of light-shielding by inclusion shells. The poorest inclusion efficiency by the γ-CD system is likely responsible for the higher relative amount of TO that can be reached by incident light, leading to the brightest orange-red color observed. On the other side, in the α-CD system the TO is mainly surrounded by light-reflecting CD shells, which reduce light absorption.

#### 3.2.2. LCSM and SEM Characterization of TO/CD Emulsions

LCSM imaging revealed clear morphological differences among freshly prepared TO/CD emulsions (Figure 3). In α- and β-CD stabilized samples, TO was dispersed into spherical droplets, of the average diameter of 27 ± 13 μm, uniformly colored in red because of the auto-fluorescence emission of carotenoids (Figure 3a,d). Densely packed layers of microcrystals appeared to self-assemble into a continuous coating at the oil/water interface forming a protective shell preventing oil droplet coalescence and giving them a rough surface (Figure 3b,c,e,f). Similar micrometric crystallized shells (cyclodextrinosomes) have been described by many authors in Pickering emulsions of different CDs with mineral (liquid paraffin, n-alkanes), synthetic (silicone, squalane), vegetable (castor, colza, olive, rice bran, soybean, wheat germ), and fish oils, as well as high purity monocarboxylic acids (C_3_-C_12_) [34,37,47,48,49,50,51]. In these systems, the spontaneous assembly and irreversible adsorption of solid particles at the oil/water interface was attributed to the in situ formation and crystallization of CD inclusion complexes with specific oil constituents (e.g., fatty acid chains of triglycerides, tetradecane, or long polymers of suitable geometry) [37,52,53]. Flexible anisotropic particles with high aspect ratios (needle or rod-shaped) were reported to be particularly efficient in stabilizing the emulsion by increasing the effective droplet coverage and intertwining when adsorbed at liquid interfaces, thus out-performing the equivalent spherical solid particles [54].

Much smaller (average Ø = ~9 ± 4 µm) fluorescent droplets with a smooth surface, indicative of a “naked” nature or an extremely fine particle coating, were instead widespread in the TO/γ-CD emulsions (Figure 3g–i). Few spherical or oblong rough TO droplets with a broad size distribution from 2.0 up to 40 µm were also occasionally observed (insert Figure 3i). Studying the formation of tetradecane/β-CD stabilized emulsions, Mathapa and Paunov [37] reported that the frequency of droplets with a smooth surface increases with increasing the oil volume fraction (ɸ_0_) and that, at ɸ_0_ = 0.1, all observed droplets had a rough surface. Thus, the overall prevalence of smooth droplets found in TO/γ-CD emulsions despite the very low ɸ_0_ (0.05) used in this study, is indicative of a different aptitude of CDs to form inclusion complexes with the TO constituents and/or in their capability to adsorb and crystallize at the oil/water interface.

According to confocal observations, SEM imaging of air-dried TO/α-CD emulsions (Figure 4a–c) revealed the presence of characteristic flat hexagonal microcrystals (Ø ≈ 2.5 µm; thickness ≈ 146 nm) forming a layered shell around TO droplets. These particles (platelets) were absent in the air-dried α-CD solution, where small triangular crystals were observed, instead (Figure 4d), corroborating the assumption that platelets arise from crystallization of inclusion complexes of α-CD with specific TO constituents. Furthermore, platelets showed different spatial orientations. Most lie with the flat side parallel to the oil/water interface, while others appear to be randomly incident, even perpendicularly, interpenetrate each other. Similar hexagonal shaped nano- and micro-platelets were reported in oil/water emulsions stabilized with α-CDs and hydrophobically-modified polysaccharides (*O*-palmitoyl-dextran or *O*-palmitoyl-amylopectin). No platelets self-assembly was observed, instead, in the presence of α-CDs alone [51]. Therefore, we could speculate that SC-CO_2_ technology co-extract with TO other compounds that mimic the function of modified polysaccharides in the formation of the hexagonal platelets, such as long chain tomato epicuticular and cuticular waxes. Further research is required to validate this assumption.

An amorphous compact surface revealing the presence of underneath crystalline material in correspondence with sporadic cracks (Figure 4e–g), or a compact background of tetragonal crystals furrowed by several roughly hemispheric cavities (Figure 4i–k) were, instead, observed in air-dried TO/β- and γ-CD emulsions, respectively. In both samples, the microcrystals observed under SEM were formed independently of the presence of TO (Figure 4h,l), indicating they simply result from uncomplexed CDs crystallization.

Mathapa and Paunov [37] reported the formation of collapsed or hollow colloidosomes after gold-sputtering under high vacuum depending on the compactness of their protective shell. This treatment was in fact able to evaporate the oil core from the CD-stabilized oil/water emulsion. Thus, the amorphous surface observed in air dried TO/β-CD emulsion may likely be the result of the collapse of β-cyclodextrinosomes, while the hemispheric cavities observed in TO/γ-CD emulsion may represent the imprints of oil droplets (oil shadows) left after oil evaporation, suggestive of a weaker shell formation or a complete lack of coating, respectively. The formation of a loose layer of β-CD rich crystals around TO droplets in β-CD-stabilized emulsions accounts likely for the low density of the disperse phase and its scarce tendency to separate by sedimentation (Figure 1, middle vial). On the other side, the ability of α-CD-encapsulated TO droplets to retain their spherical shape under vacuum is consistent with the formation of highly compact and thick α-CD rich capsules responsible for a fast sedimentation rate (Figure 1, left vial). Nevertheless, crack opened hollow cyclodextrinosomes were found also in the TO/α-CD sample (Appendix A).

SEM observation of TO/α-CD powders obtained by freeze-drying confirmed the presence of stable cyclodextrinosomes interspersed into an irregular porous matrix (Figure 5a). The dehydration process seemed to affect the morphology of the platelets, which lost their original hexagonal shapes and became wider and irregular (Figure 5b). A complex network of semicrystalline material apparently embedding collapsed structures was, instead revealed by SEM imaging of the TO/β-CD powders. Finally, TO/γ-CD powder appeared organized into smooth thin sheets with pores likely produced from the formation of ice crystals during the freeze-drying process, similarly to other natural supramolecular complexes [55].

#### 3.2.3. FTIR-ATR

Figure 6 shows the FTIR-ATR spectra of crude TO, native CDs, and TO/CD powders arising from emulsion freeze-drying. The infrared spectrum of TO layered onto the IRE crystal (Figure 6, trace A) is dominated by signals of acylglycerols, accounting for at least 95% of the total weight. Absorption bands arising from oil solubilized bioactive compounds, including carotenoids, were not detectable. Main peaks in the spectrum are assigned to specific functional group vibrations arising from fatty acids and their esters, namely vinyl C-H stretch (3010 cm^−1^), asymmetric and symmetric methylene C-H stretch (2924 and 2854 cm^−1^), ester C=O stretch (1745 cm^−1^), free carboxylic acid C=O stretch (1712 cm^−1^), methylene C-H bend (1464 cm^−1^), ester C-O stretch (1238 cm^−1^), and C-C stretch (1162 cm^−1^).

Infrared peaks of CDs fall mainly in the region between 1180 and 1000 cm^−1^ and can be assigned to the stretching of glycosidic C–O–C and C–OH bonds, with the most intense peak at around 1025 cm^−1^. The typical scissoring vibration peak of H_2_O at 1640 cm^−1^, indicating the presence of hydration water, is observed in the spectra of all α-, β-, and γ-CD powders. It is well known that different crystalline forms of CDs may retain a variable number of water molecules [56]. The presence of hydration water in the CD samples is further confirmed by DSC measurements, as discussed later.

FTIR spectra of the TO/CD powders obtained from freeze-drying of α-, β-, and γ-CD stabilized emulsions present the spectral features arising from both TO and CD components. However, it is noteworthy the absence of the band at 1712 cm^−1^ in all TO/CD powder spectra, suggesting at first glance three possible scenarios: (a) free fatty acids are selectively excluded from TO/CD emulsions, (b) they undergo chemical modification upon inclusion complexes formation, (c) they are lost during the freeze-drying process. Infrared spectra show that the freeze-dried powder obtained from TO/α-CD emulsion presents the higher oil content. The band at 1745 cm^−1^, representative of TO oil esters, is indeed more intense than those observed in TO/β-CD and TO/γ-CD powder spectra, taking the relevant CD band at 1025 cm^−1^ as reference signal. This trend is well depicted in the inset of Figure 6 showing the ratio between marker signals of CD and TO. It should be taken into account that the original emulsions were prepared with CD solutions at the same initial concentration (10 mM); therefore, the lower intensity ratio between marker IR bands of CD and TO in TO/α-CD sample could be partially due to the lower number of glucose units in α-CD structure. Nevertheless, we argue that the noteworthy high oil content in TO/α-CD powder, detected by infrared spectroscopy, is mainly ascribable to the higher stability of α-CD-capsules, which better resist to the freeze-drying process avoiding any loss of encapsulated oil during the sublimation step. This interpretation of IR data agrees with SEM micrographs of powders (Figure 5) showing that intact round-shaped cyclodextrinosomes are detectable only in TO/α-CD powder. Even though freeze-drying does not involve pressures as low as those employed during SEM measurements, a certain loss of liquid components is indeed expected. To further investigate this aspect, gaining information on the chemical composition of suprastructures formed during the CD-driven emulsification process, we acquired the infrared spectrum of the sole encapsulated material. For this purpose, CD-coated oil droplets were separated from the continuous aqueous phase by a centrifugation step, washed four times with pure water, cast onto the ATR crystal, and air-dried. Relevant spectra, normalized to the reference CD signal at around 1030 cm^−1^, are shown in Figure 7 as acquired, without applying any intensity normalization procedure. Interestingly, in this case all three CDs behave in a similar fashion with the pellet showing a higher oil content compared to the corresponding freeze-dried powder and an almost constant ratio of the intensity of marker IR bands of sugar (1030 cm^−1^) and oil (1745 cm^−1^) components (see histogram in the inset of Figure 7). A slight hypsochromic shift of the marker CD band with respect to powders was observed. It is noteworthy the presence in these spectra of the shoulder at 1712 cm^−1^, indicative of encapsulated free fatty acids, supporting the hypothesis that these oil components are mainly lost during the freeze-drying process and are instead retained in air-dried samples. The slightly higher ratio between glucose units and fatty acid ester groups in β-CD-encapsulated oil sample is indicative of a higher amount of β-cyclodextrin involved in the formation of the oil droplet shells. Unlike α-CD and γ-CD decorated oil droplets, which present a smooth outline in LCSM images (Figure 3c,i), β-CD-encapsulated droplets show a characteristic spongy appearance (Figure 3f), which likely accounts for the poor compactness and breaking strength of β-cyclodextrinosomes, despite the higher number of β-cyclodextrin molecules involved. The high β-CD relative content enlightened by the IR spectrum of sedimented β-CD encapsulated oil agrees with quantitative data provided by the infrared absorption of corresponding supernatants (Appendix A). The amount of uncomplexed CD in the continuous phase of α-, β-, and γ-CD emulsions is indeed 81%, 67%, and 94%, respectively. The very low γ-CD amount (≈6%) removed from the continuous phase during the emulsification process can be explained with the low TO encapsulation yield. In fact, even with an oil/CD/ratio comparable to that of TO/α-CD pellet (Figure 7), TO/γ-CD sedimented phase is quantitatively very scarce and a consistent layer of unencapsulated oil appears on the top of the tube after phase separation (Figure 1, right vial).

#### 3.2.4. DSC

DSC scans were run to provide information on the physical and chemical processes occurring between the different CDs and TO on heating, by comparing the thermal behavior of the single components and that of their TO/CD powders. This technique, in fact, is recognized to be a useful tool to study interactions between drug compounds and CDs in solid state and to identify the formation of inclusion complexes with CDs, since the peak temperatures of melting or decomposition processes of the guest molecules are likely to change, or these phenomena can even disappear, when they are included into CD cavities [57,58,59,60]. The shifting of the different peak temperatures (i.e., dehydration, degradation, etc.) found in the DSC curves indicates, therefore, probable interaction between CD and guest molecule, which, in turn, causes more stable inclusion complex formation [60,61]. By comparing the DSC thermograms of the individual components with those of the TO/CD powders it is, then, possible to get information about solid-state modifications and interactions between the components and confirm or exclude complexation [62].

Figure 8a shows the DSC thermograms of α-CD and TO/α-CD powder. According to Hădărugă et al. [63] the large endothermic area displayed by α-CD up to around 160 °C corresponds to the loss of water. The different peaks observed in the thermogram (at around 80, 115, and 141 °C) are related to various physically bonded water molecules (i.e., “surface water” and “strongly-retained water”). The total absorbed heat due to water dissociation from α-CD hydrate is about 344 J/g, this value being consistent to that reported in literature (i.e., 338 J/g). The smaller endothermic phenomena noticed at higher temperatures, with a peak at 224 °C, can be related to transitions of α-CD from a form to another, with possibly rearrangement of the crystal structure after the loss of water molecules Hădărugă et al. [63]. The thermal decomposition of α-CD is reported to start at temperatures greater than 270–280 °C and it is completed around 400 °C [63].

The DSC thermogram of the crude TO (Figure 8) suggests that no thermal phenomena occur in the range of temperatures of interest, i.e., up to 250 °C. At this temperature, the onset of thermal degradation is noticed.

DSC thermogram obtained for TO/α-CD powder presents a single large endothermic peak, centered at about 92 °C. The much lower intensity of this peak with respect to α-CD and its different shape suggest the presence of a low amount of water, less strongly bound. No other peaks were noticed in the sample. Different dehydration mechanisms can be, therefore, hypothesized for free α-CD and its combination with TO. Moreover, the small endothermic peak, attributed to the transition of α-CD from a form to another, disappeared in TO/α-CD powders sample. As previously highlighted, the disappearance/shifting of the peaks detected for α-CD are an indirect proof that some interactions took place between TO and α-CD [64,65]. The onset of degradation is registered beyond 270–280 °C, suggesting that the thermal stability of TO is enhanced in presence of α-CDs. An increased decomposition temperature can be regarded as an additional evidence of the, at least partial, formation of inclusion complexes [62]. The calorimetric observations are, therefore, in accordance with what observed at confocal SEM and FTIR analysis.

Many studies on thermal behavior of β-CD and its complexes have been performed since the latter are particularly appropriate for pharmaceutical formulations and very cheap. DSC analysis performed on β-CDs revealed a single large endothermic peak centered at around 123 °C (Figure 8b), corresponding to dehydration and release of water molecules from the cavity, as reported by Hădărugă et al. [63]. Unlike α-CDs, no specific peaks for “surface” and “strongly-retained” water molecules have been observed in this case [63,66]. The enthalpy of the endothermic phenomenon is on average 413 J/g, again in line with literature (i.e., 401 J/g). At higher temperatures, a phase transition of the β-CD is revealed by exothermic and endothermic small peaks, centered at around 220 and 227 °C, respectively. These exo-endothermic effects, observed also by Guimarães et al. [67], have been proven to be due to structural modifications of anhydrous β-CD crystal structure [63]. The beginning of degradation step is not visible in our thermogram since it is reported to start beyond 300 °C [63].

A single endothermic peak can be observed also for the TO/β-CD powder (Figure 8b). It is centered at 111 °C, i.e., at a temperature about 10 °C lower than that measured on pristine β-CD; the intensity of this peak is again much lower than that observed for β-CD, while the shapes are quite similar. The TO/β-CD powder is likely to contain less and mostly superficial water molecules that can be more easily removed, therefore, the temperature of the endothermic peak and its intensity are both reduced [68,69]. In addition, TO/β-CD powder still shows a small endothermic peak, which is smaller and centered at a lower temperature (i.e., 218 °C) with respect to free β-CD; the exothermic one, besides, disappeared. All the previous observations can be reasonably regarded as evidence of the low interactions existing between TO molecules and β-CDs [70]. Once again, these results represent an indirect confirmation of what was found in the other analyses. Finally, degradation starts beyond 270–280 °C, i.e., at temperatures greater than that measured for crude TO, thus proving the improved thermal stability of the system.

DSC thermograms of γ-CDs (Figure 8c) showed a large peak relative to water release at about 110 °C. Although this temperature was much higher than the 81 °C found for hydrate γ-CDs, peak shape was similar to previous reports [63] and displayed, again, a sort of sum of peaks corresponding to different types of water molecules, even though not so marked as in α-CDs. The heat absorbed during water release from hydrate γ-CDs was around 290 J/g, not so different from that reported in literature (i.e., around 330 J/g). The decomposition of γ-CD starts above 270 °C, as previously reported [63]. No other thermal phenomena were observed between water release and degradation processes.

A single endothermic peak was observed also for TO/γ-CD powder (Figure 8c), with a shape quite similar to that observed for γ-CD and an intensity slightly smaller. The peak was centered at about 92 °C, i.e., at a temperature much lower than that measured on free γ-CD. This observation suggests that residual physically bonded “surface water” molecules are mainly present in TO/γ-CD powder. No other peak is observed for such system before the start of degradation, as for free γ-CDs. Thermal degradation, occurring at around 280 °C, confirms the achievement of a good thermal stability also for TO/γ-CD powder. The previous observations suggest that a basically physical combination of the two single components, i.e., γ-CD-TO, took place, confirming the observations from SEM and FTIR investigations.

### 3.3. Antioxidant Activity of the TO/CD Emulsions and Powders

In general, assessing the transfer of antioxidant equivalents from a hydrophobic environment to a water dispersed phase is not a simple task. Main complications arise from the different assay conditions employed, which require different standards, i.e., α-tocopherol and Trolox for hydrophobic and hydrophilic environment, respectively. This means that the direct comparison of α-TEAC and TEAC values, obtained for hexane-solubilized and water-dispersed hydrophobic antioxidants, respectively, provides only a semiquantitative analysis of antioxidant capacity transfer yield from one phase to another. Moreover, the TEAC value of supramolecular and colloidal systems does not depend solely on the amount of encapsulated antioxidant compounds, but also on the actual accessibility of the aqueous radical probe ABTS^•+^ to the hydrophobic antioxidants dispersed in water by the specific emulsifying agent or supramolecular carrier. Nevertheless, the assessment of apparent TEAC of emulsions and micro-heterogeneous systems containing hydrophobic antioxidants provides valuable information about both encapsulation yield and actual ability of encapsulated antioxidants to intercept and neutralize aqueous radicals, such as ABTS^•+^. A further issue concerns the kinetics of radical neutralization, which can be slowed down in a micro-heterogeneous environment affecting TEAC values in a time-dependent way. Although, the time course of the ABTS^•+^ decolorization reaction of all assayed representative samples (crude TO, TO/α, β-CD powders, and TO/α-CD emulsion) showed a linear trend, differences in the slope of the interpolation lines were observed, indicative of either a slight dependence on CD typology or sample physical state of antioxidant kinetics (Appendix A).

The α-Tocopherol Equivalent Antioxidant Capacity (α-TEAC) value of crude TO dissolved in hexane was 64 ± 3 µEq/g, in rough agreement with the value of 86 µEq/g calculated on the basis of carotene and lycopene content.

The TEAC values of TO/α-CD and TO/β-CD emulsions, referred to the mass of loaded TO, were slightly higher than that of the crude TO, while a lower value was registered for TO/γ-CD (Table 2). This finding agrees with the higher efficiency of α- and β-CDs as emulsifying agents for TO compared to γ-CD. Moreover, the TEAC values of emulsions, equivalent to the α-TEAC of crude TO, indicated that the encapsulated hydrophobic antioxidants retain their ability to scavenge ABTS^•+^. TEAC values of suspended TO/CD powders, referred to the mass of freeze-dried sample (Table 2), were influenced by the molecular mass of specific CDs (employed at the same molar concentration in the starting emulsion) and by a possible loss of oil components during the freeze-drying process, which may affect the TO/CD mass ratios. However, the TEAC values of powders referred to the mass of loaded TO were also calculated for a more direct data comparison, considering the different CD molecular weights and neglecting any possible change of TO/CD mass ratio after emulsion freeze-drying (Table 2). These data indicate that the freeze-drying step does not result in a loss of TO antioxidant capacity, which instead appears highly stable. In the case of α-CDs and γ-CDs a slight increase of TEAC was even observed moving from the emulsion to the relevant powder. Moreover, the TEAC values of the freeze-dried powders confirmed the better performances of α- and β-CDs with respect to γ-CDs not only as oil-dispersing agents, but also as carrier systems for the vehiculation of active hydrophobic antioxidants across aqueous media. Furthermore, the measured TEAC values were in line with the antioxidant EE%, indicating that the loss of acylglycerols occurring during freeze-drying, as revealed by FTIR spectra, was not accompanied by a significant loss of oil dispersed antioxidants.

### 3.4. Effect of Storage Conditions on Carotenoid Shelf-Life Stability and Degradation Kinetics

Carotenoids retention during storage of TO, TO/CD emulsions, and powders is reported in Figure 9 and Figure 10. At 25 °C, a rapid decline of β-carotene and total [*Z*]-lycopene isomers occurred in all samples. In the short period (30 and 15 days storage for emulsions and powders, respectively), CD encapsulation significantly increased β-carotene stability compared to crude TO. Afterwards, samples showed diverse behavior. β-Carotene was almost undetectable in both TO/α- and β-CD emulsions after 60 days of storage, but partially retained (26% and 22%, respectively) in TO/γ-CD emulsion and TO, even at the end of storage (90 days). Degradation of β-carotene proceeded even faster in the powders and was almost complete after 45, 60, and 90 days storage in TO/β-, α-, and γ-CD samples, respectively. Similarly, total [*Z*]-lycopene isomers quickly degraded between 0 and 30–45 days in all samples, reaching a plateau in the subsequent period. At 90 days storage, retention of total [*Z*]-lycopene isomers was approx. 20%, 3%, and 9% in TO/γ-CD emulsion, TO/γ-CD powder, and TO, respectively. The content of the [*Z*]-isoforms in TO/γ-CD emulsions and powders was generally higher than in TO/α-CD and TO/β-CD samples. Interestingly, a peak, tentatively identified as 5-[*Z*]-lycopene on the basis of literature data [71], appeared in the chromatograms of TO/γ-CD samples after 30 days storage and increased over time contributing up to 57% and 73% of the total identified [*Z*]-lycopene isomers at the end of storage period, in the emulsion and powder, respectively. Regardless of CD type, *all*-[*E*]-lycopene was less stable in the emulsions and powders than in TO. Nevertheless, no significant differences were found between *all*-[*E*]-lycopene retention in TO and TO/γ-CD emulsion in the first 15 days storage.

Low storage temperature (4 °C) slowed down carotenoid degradation. Retention of β-carotene in the emulsions and powders was higher than in crude TO. These results are in agreement with Przybysz et al. [72] who emphasized that the retention of α- and β-carotene in Arabic gum and maltodextrin microspheres was higher than the retention in a crude oil solution. After 90 days, 91% of β-carotene was retained by the TO/β-CD emulsion, followed by TO/α- (85%) and γ-CD (61%) emulsions. The TO/CD powders provided reduced protection compared to the corresponding emulsions with retention values comprised between 56% and 77%. Furthermore, in crude TO, β-carotene degradation was faster in the first month at both 25 and 4 °C than in the remaining storage period.

Similarly, Przybysz et al. [72] reported that the degradation of carotenes in oil solutions was faster in the first weeks of the storage. [*Z*]-Lycopene isomers appeared better protected in TO/α-, β-, and γ-CD emulsions than TO. In TO/β-CD emulsion, the total content of [*Z*]-lycopene isomers decreased rapidly (up to 30 days) and remained almost stable (68–57%) in the following storage period. Total [*Z*]-lycopene isomers were less stable in the powders than in the emulsion. After 90 days storage, their retention values were 69% and 30% in TO/γ- and α-CD powders, respectively, and 14% in TO/β-CD emulsion, this was lower than crude TO. Again, a putative 5-[*Z*]-lycopene isomer was observed exclusively in the TO/γ-CD samples after 30 days storage contributing from 3% to 27% of the total identified [*Z*]-lycopene isomers. Regardless of temperature, *all*-[*E*]-lycopene retention was higher in crude TO than in the other samples. More generally, our data indicated that carotenoid stability over time (90 days storage) was lower in the powders than in the emulsions. The porous structure of the powder may favor oxygen diffusion within the sample negatively affecting carotenoids shelf-life stability [73,74,75].

Table 3 reports the correlation coefficient (r^2^), the degradation rate constant (k), and half-life time (t_1/2_) values of carotenoids in TO, TO/CD emulsions, and powders stored at 25 and 4 °C, in the dark.

Based on the r^2^, degradation of carotenoids followed a first order kinetic in all samples. Kinetic parameters were strongly influenced by storage temperature. Indeed, at 4 °C, lower k and higher t_1/2_ values were obtained compared to 25 °C. Recently, Amiri-Rigi and Abbasi [76] concluded that at 5 °C TO/water microemulsions prepared using saponins as natural surfactants can stay almost unchanged up to 90 days.

At 25 °C, t_1/2_ values of β-carotene were substantially lower than those obtained using maltodextrins (50–431 days) or arabic gum/maltodextrins (64 days) as encapsulating agents [73,77,78]. In TO/CD emulsions and powders, t_1/2_ for lycopene ranged between 11 and 42 days, values similar or higher than those reported by Xue et al. [79] for TO encapsulated in zein by spray-drying (11 days at 25 °C) and by Chiu et al. [80] for SC-CO_2_ extracted oil from tomato pomace encapsulated in poly-γ-glutamic acid (25 days at 25 °C). Furthermore, Robert et al. [81] reported t_1/2_ values of 5–23 days (at 21 °C) for β-carotene and *all*-[*E*]-lycopene from Rosa Mosqueta (*Rosa rubiginosa* L.) oil encapsulated in starch and gelatin matrices.

At 4 °C, carotenoid degradation rate in crude TO was [*Z*]-lycopene isomers > β-carotene > *all*-[*E*]-lycopene, while in the TO/CD emulsions and powders it was *all*-[*E*]-lycopene > [*Z*]-lycopene isomers > β-carotene in accordance with our previous report on the TO/α-CD powder [9]. TO/β-CD emulsions exhibited the highest t1/2 values for β-carotene (608 days), [*Z*]-lycopene isomers (117 days), and *all*-[*E*]-lycopene (38 days), but after freeze-drying the best t_1/2_ values were found in the TO/γ-CD powder (β-carotene (205 days), [*Z*]-lycopene (146 days) and *all*-[*E*]-lycopene (34 days)). However, in TO/CD samples, the t_1/2_ values of β-carotene and lycopene were lower than those obtained by Kha et al. [82] for Gac (*Momordica cochinchinensis* (Lour.) Spreng.) oil encapsulated in whey protein concentrate and Arabic gum at −20 and 10 °C (β-carotene between 2310 and 710 days; lycopene, between 1733 and 1155 days).

## 4. Conclusions

Taken together, the data indicate substantial differences on the aptitude of CD types to form supramolecular complexes with TO, as well as on their size and morphology and suggest the following conclusions: (a) α-CDs show satisfactory oil EE%, promoting the formation of compact and hard shells around disperse oil-droplets, conferring fast sedimentation rate to cyclodextrinosomes and resistance to dehydration; (b) β-CDs show the best oil dispersion, forming β-CD decorated oil spheres with a high CD content and a rough surface, which proved to sediment slowly and loose stability in the dehydrated state; (c) γ-CDs present the lowest oil dispersion, with the formation of a small amount of cyclodextrinosomes, characterized by a CD/TO ratio similar to that of α-CD; (d) based on the antioxidant activity of emulsions and powders, α- and β-CDs revealed better performances than γ-CDs as oil-dispersing agents and carrier systems of hydrophobic antioxidants across aqueous media.

The time course stability of TO/CD emulsions and powders evidenced a rapid decline of carotenoid content at 25 °C, regardless of CDs type. Low temperature slowed down β-carotene degradation in both TO/CD emulsions and powders, while its effect on lycopene isomers was negligible. Any type of CDs seemed suitable to improve *all*-[*E*]-lycopene shelf-life stability.

Based on the overall characteristics, TO/CD emulsions and powders could be used as high-quality solvent/surfactant free quite stable ingredients in the preparation of nutraceutical, cosmeceutical, and/or pharmaceutical products.

## Figures and Tables

**Figure 1 foods-09-01553-f001:**
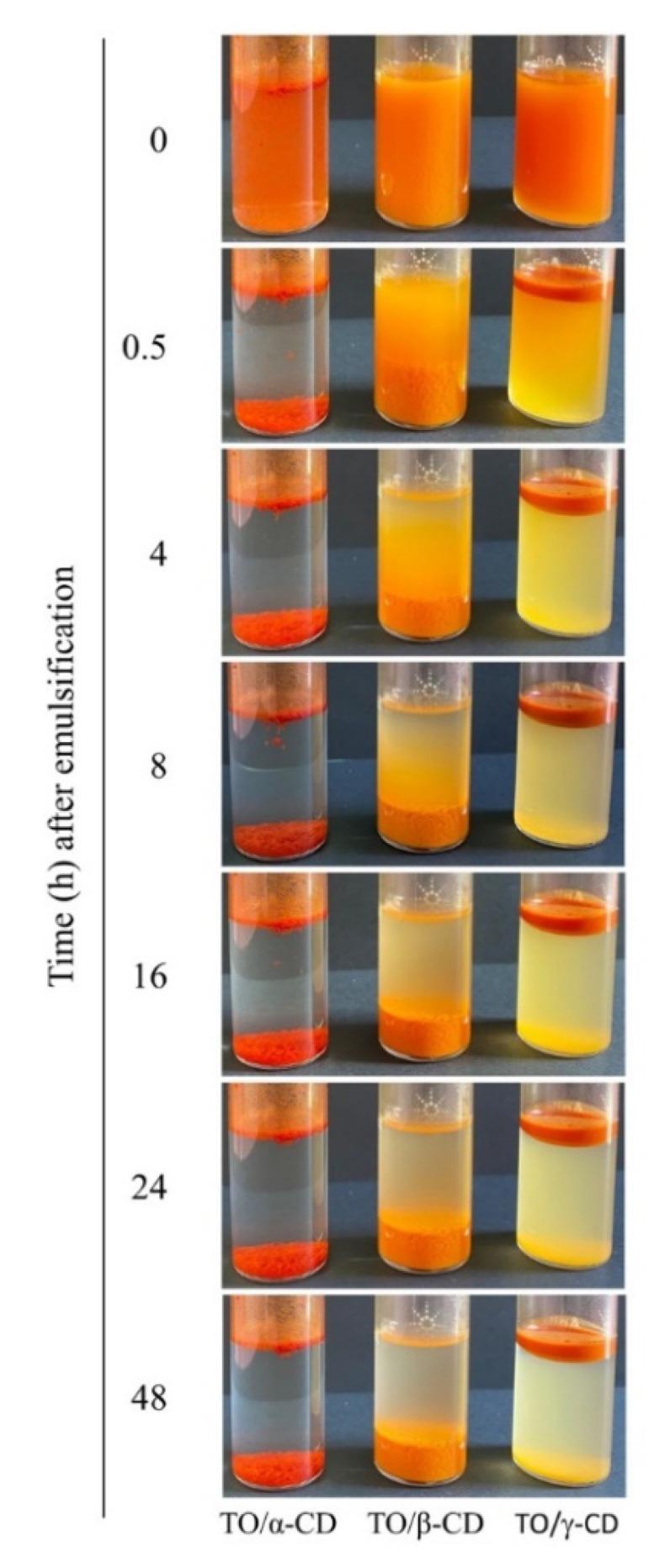
Macroscopic aspect of TO (tomato oil)/α-, β-, and γ-CD (cyclodextrin) emulsions over time.

**Figure 2 foods-09-01553-f002:**
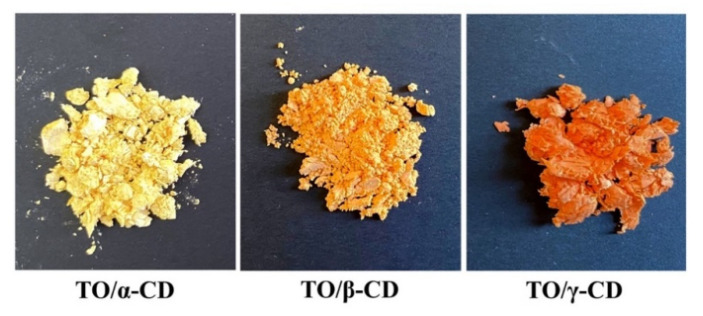
Macroscopic aspect of TO/α-, β-, and γ-CD powders.

**Figure 3 foods-09-01553-f003:**
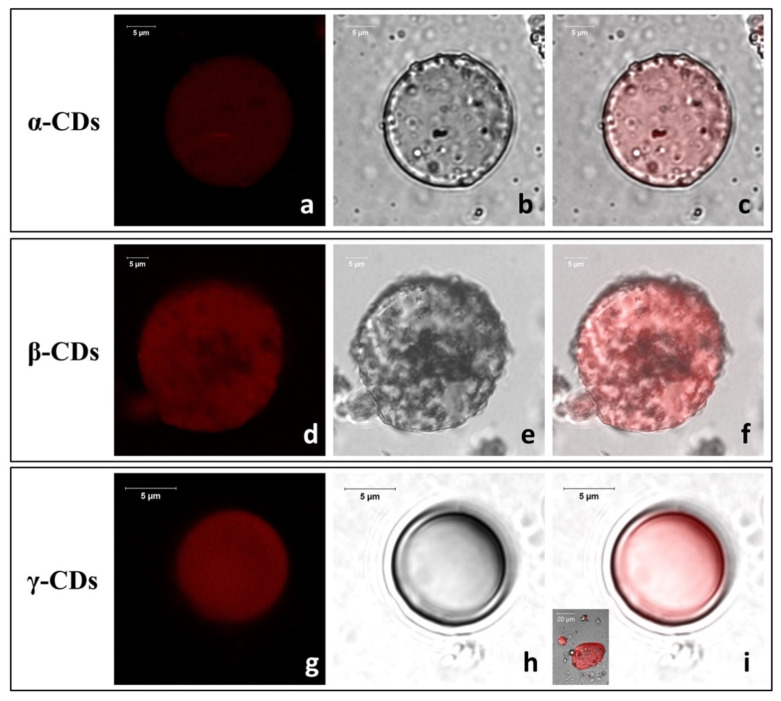
Representative confocal micrographs (fluorescence (**a**,**d**,**g**), optical (**b**,**e**,**h**), and merge images (**c**,**f**,**i**)) of freshly prepared TO/CD emulsions.

**Figure 4 foods-09-01553-f004:**
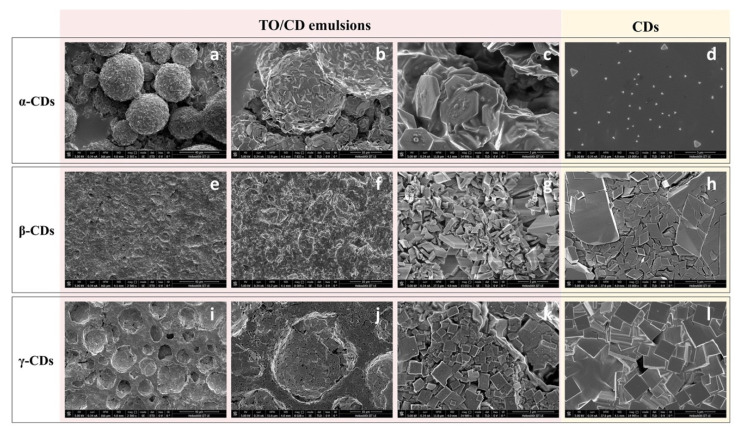
Scanning electron micrographs of air-dried TO/CD emulsions (**a**–**c**, TO/α-CD; **e**–**g**, TO/β-CD; **i**–**k**, TO/γ-CD) and CD solutions (**d**, α-CD; **h**, β-CD, **l**, γ-CD) at different magnification.

**Figure 5 foods-09-01553-f005:**
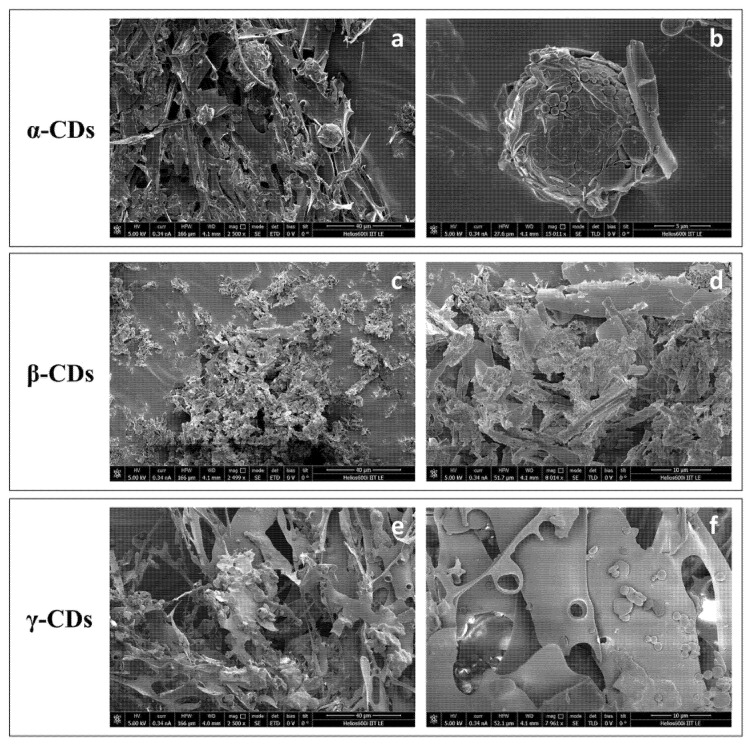
SEM micrographs of TO/CD powders (**a**,**b**, TO/α-CD; **c**,**d**, TO/β-CD; **e**,**f**, TO/γ-CD) at different magnification.

**Figure 6 foods-09-01553-f006:**
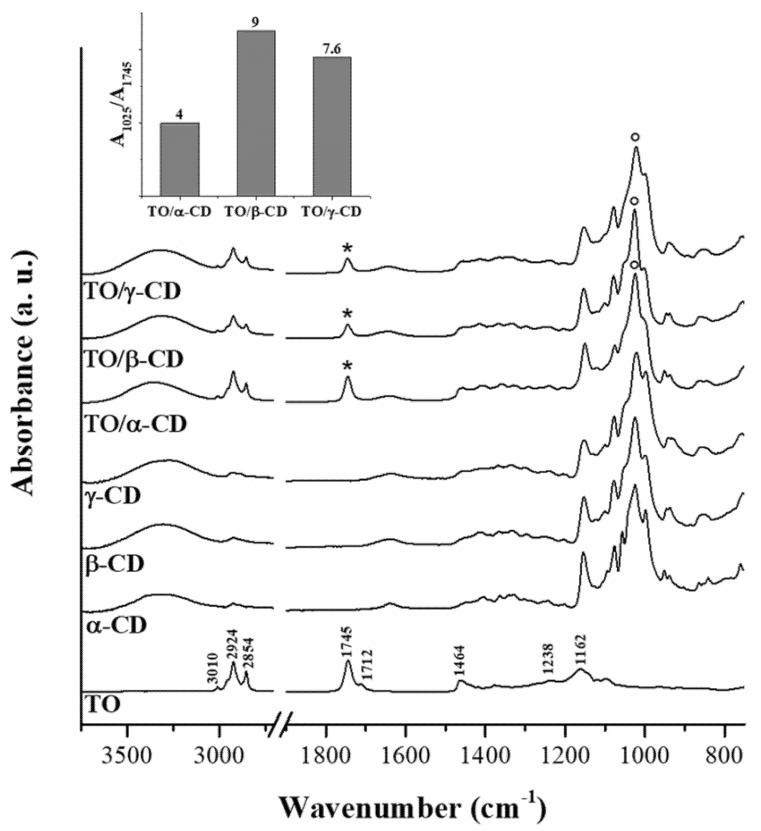
FTIR-ATR (Fourier transform infrared-attenuated total reflection) spectra of TO, CD native powders, and TO/CD freeze-dried powders. When present, the CD most intense band at around 1025 cm^−1^ was taken as reference signal for normalization of traces. Asterisks and circles indicate marker bands for TO and CD, respectively. The trend of their ratio in TO/CD freeze-dried powders is depicted in the histogram.

**Figure 7 foods-09-01553-f007:**
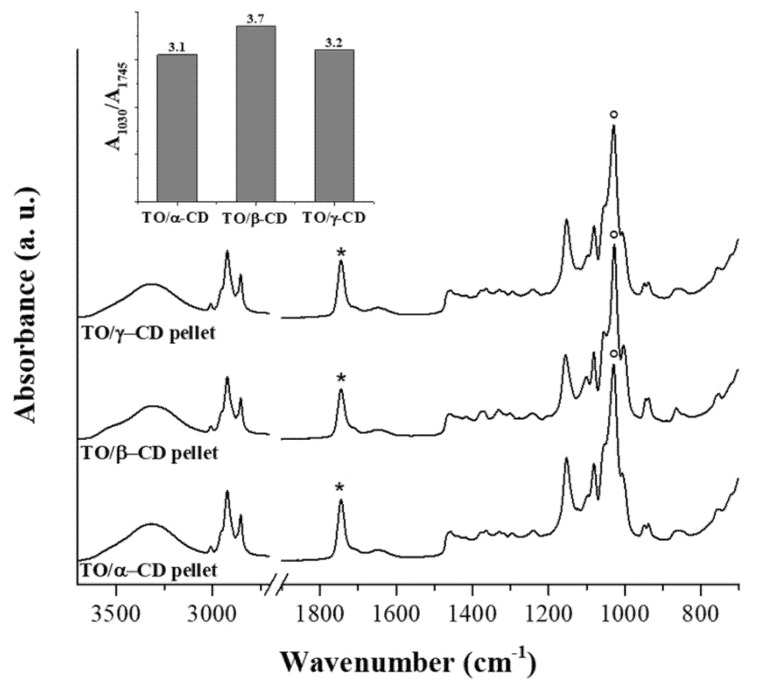
FTIR-ATR spectra of CD-oil particles sedimented from TO/α-CD, TO/β-CD, and TO/γ-CD emulsions. Samples were deposited by casting/evaporation technique (see text for details). The CD most intense band at around 1030 cm^−1^ was taken as reference signal for normalization of traces. Asterisks and circles indicate marker bands for TO and CD, respectively. The trend of their ratio in TO/CD particles is depicted in the histogram.

**Figure 8 foods-09-01553-f008:**
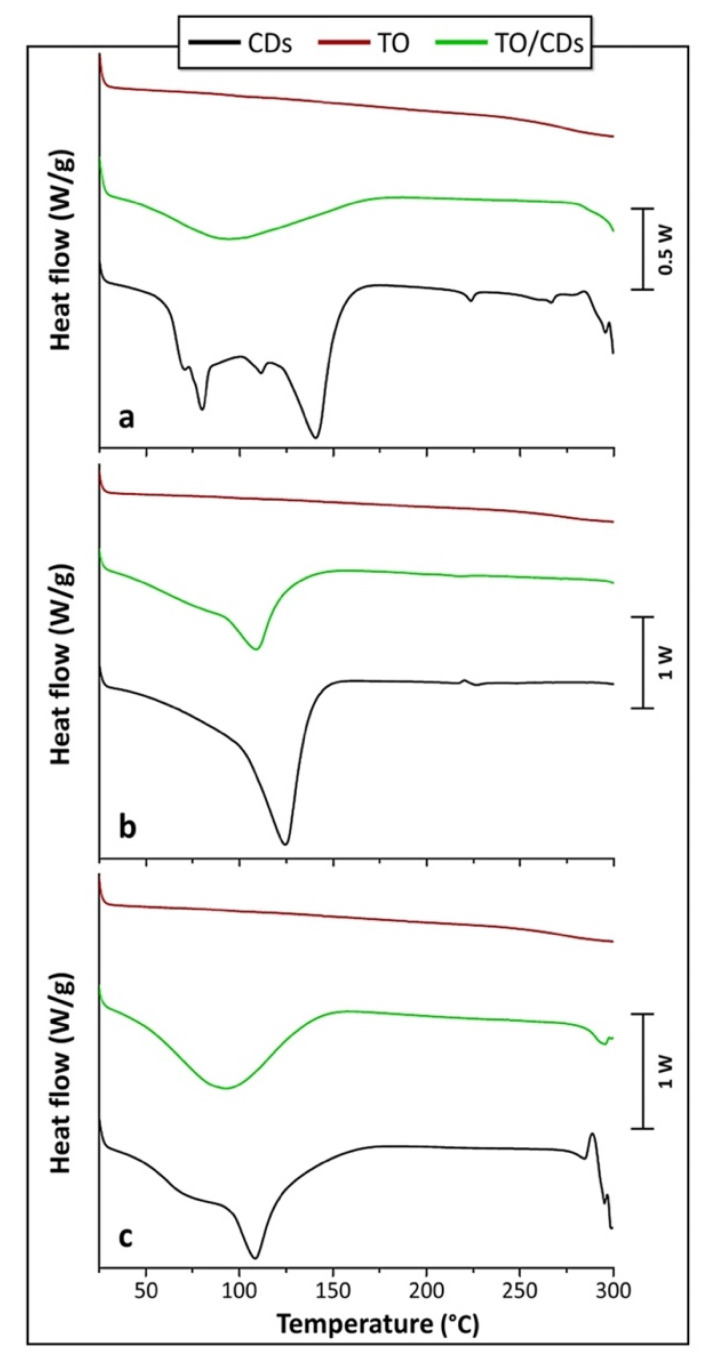
DSC (differential scanning calorimetry) curves of (**a**) α-CDs, TO, and TO/α-CD powder; (**b**) β-CDs, TO, and TO/β-CD powder; (**c**) γ-CDs, TO, and TO/γ-CD powder.

**Figure 9 foods-09-01553-f009:**
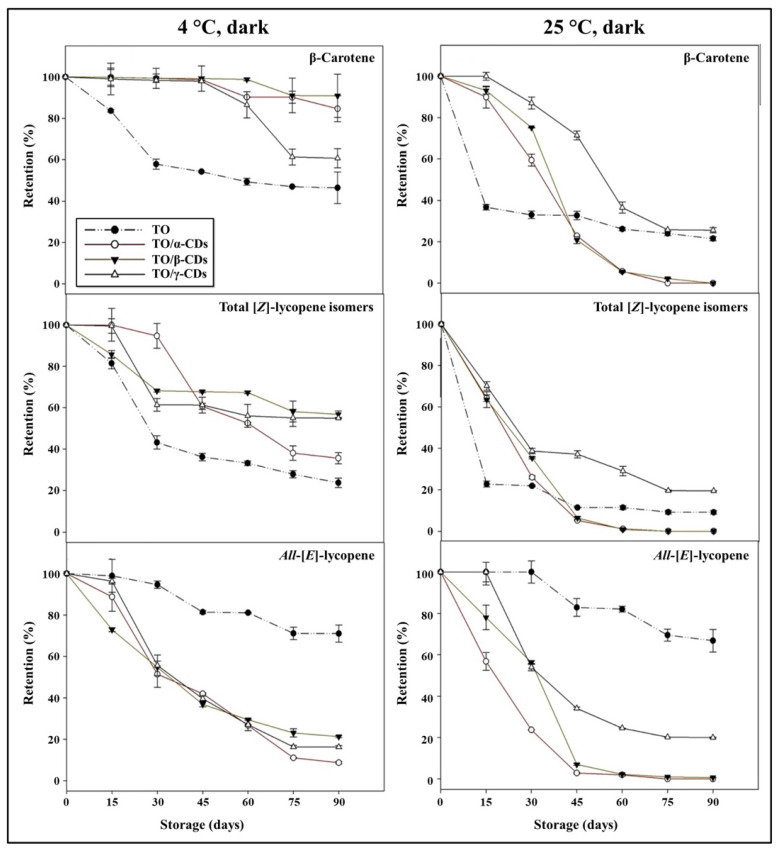
Shelf-life of carotenoids in TO and TO/α-, βm- and γ-CD emulsions stored at 4 and 25 °C in the dark.

**Figure 10 foods-09-01553-f010:**
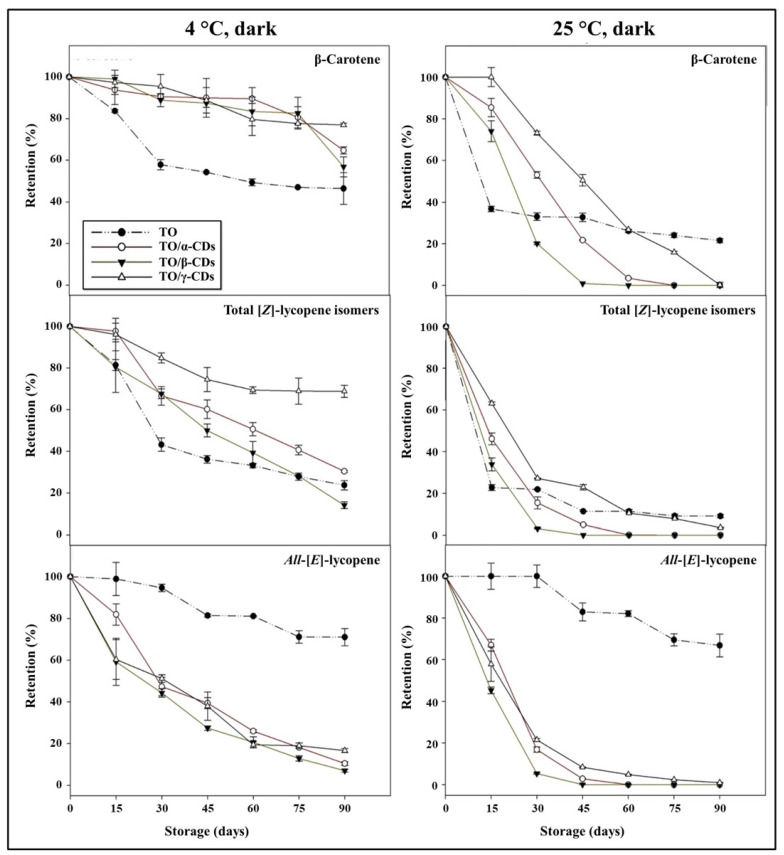
Shelf-life of carotenoids in TO and TO/α-, β-, and γ-CD powders stored at 4 and 25 °C in the dark.

**Table 1 foods-09-01553-t001:** Carotenoid composition of tomato SC-CO_2_ extracted oleoresin (TO) and TO/α-, β-, and γ-CD powders and entrapment efficiency (EE%). Different superscript letters denote significant differences among samples for each row (Tukey post hoc test, *p* < 0.05).

Carotenoids	TO	Powders		
		TO/α-CDs	TO/β-CDs	TO/γ-CDs
	mg/100 g	EE% (mg/100 g Oil)	EE% (mg/100 g Oil)	EE% (mg/100 g Oil)
β-carotene	34.1 ± 2.3 ^a^	96.8 (33.0 ± 1.2 ^a^)	97.4 (33.2 ± 1.2 ^a^)	68 (23.2 ± 1.7 ^b^)
[*Z*]-lycopene isomers	229.2 ± 2.8 ^a^	76.5 (175.3 ± 3.2 ^b^)	71.5 (163.9 ± 2.9 ^c^)	42 (96.1 ± 2.3 ^d^)
*All*-[*E*]-lycopene	615.4 ± 31.1 ^a^	54.0 (332.3 ± 16.2 ^b^)	57.0 (350.8 ± 12.2 ^b^)	44 (270.9 ± 6.5 ^c^)
Total	878.7 ± 35.5 ^a^	61.5 (540.2 ± 10.6 ^b^)	62.4 (547.9 ± 16.3 ^b^)	44 (390.2 ± 10.5 ^c^)

**Table 2 foods-09-01553-t002:** Antioxidant activity of TO, and TO/α-, β-, and γ-CD powders evaluated by TEAC assay. The antioxidant activity of TO is given as α-Tocopherol Equivalents per mass of oil; for TO/CD emulsions and powders it is expressed as Trolox Equivalents per mass of loaded oil or total powder. Different lowercase and capital letters denote significant differences among samples expressed per mass of loaded oil and total powder, respectively (Tukey post hoc test, *p* < 0.05).

Samples	TEAC µEq/g TO (µEq/g Powder)
TO	64 ± 3 ^bc^
Emulsions	
TO/α-CDs	75 ± 5 ^ab^
TO/β-CDs	73 ± 5 ^ab^
TO/γ-CDs	54 ± 4 ^c^
Powders	
TO/α-CDs	85 ± 9 ^a^ (29 ± 3 ^A^)
TO/β-CDs	72 ± 6 ^ab^ (22 ± 2 ^B^)
TO/γ-CDs	68 ± 6 ^bc^ (19 ± 2 ^B^)

**Table 3 foods-09-01553-t003:** Correlation coefficient (r^2^), degradation rate constant (k), and half-life time (t_1/2_) values of carotenoids in TO, TO/CD emulsions and powders stored at 25 and 4 °C, in the dark.

Samples	Temp (°C)	β-Carotene			[*Z*]-Lycopene Isomers			*All*-[*E*]-Lycopene		
		r^2^	k (10^−3^ d^−1^)	t_1/2_ (d)	r^2^	k (10^−3^ d^−1^)	t_1/2_ (d)	r^2^	k (10^−3^ d^−1^)	t_1/2_ (d)
TO	25	0.74	13.5	51	0.76	23.0	30	0.91	5.1	136
	4	0.85	11.9	58	0.92	16.0	43	0.94	4.4	158
Emulsions										
TO/α-CDs	25	0.93	59.9	12	0.92	60.4	11	0.93	37.7	18
	4	0.85	1.9	365	0.95	13.2	52	0.96	28.3	24
TO/β-CDs	25	0.94	56.8	12	0.90	61.5	11	0.98	18.0	38
	4	0.71	1.1	608	0.89	5.9	117	0.93	16.5	42
TO/γ-CDs	25	0.91	18.2	38	0.94	18.5	37	0.93	21.0	33
	4	0.77	6.1	113	0.76	33.9	21	0.96	23.3	30
Powders										
TO/α-CD	25	0.93	60.6	11	0.93	23.1	30	0.90	58	12
	4	0.77	3.8	180	0.97	13.3	52	0.99	24.8	28
TO/β-CD	25	0.81	56.9	12	0.76	62	11	0.80	64	11
	4	0.75	5.1	137	0.95	20.2	34	0.99	28.1	25
TO/γ-CD	25	0.77	28.2	25	0.99	18.5	36	0.99	21.0	33
	4	0.94	3.4	205	0.93	4.7	146	0.95	20.6	34

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
