# Peer review of "Tomato Oil Encapsulation by α-, β-, and γ-Cyclodextrins: A Comparative Study on the Formation of Supramolecular Structures, Antioxidant Activity, and Carotenoid Stability"

_foods, 2020, doi:10.3390/foods9111553_

Round 1

Reviewer 1 Report

The article by Durante et al presents an interesting encapsulation of tomato oil in Cyclodextrins. In general terms the work has enough quality for foods and is well-conducted. However, some points should be addressed before:

  1. Introduction: Why authors choose CDs? Please cite the potential applications of CDs in foods.
  2. Point 2.6 and results. I strongly recommend to test the stability in a simple food matrix, maybe the called "food model" could be a good option.
  3. L134: Please define BHT
  4. Figure 2. It is interesting the batocromic displacement, why has it been? Did the authors have some hyphothesis?
  5. Figure 3-5, please improve the caption. Specify each letter "a, b,c..."
  6. Antioxidant activity: It should be better explained: 1) How did authors normalize the quantity of TO added? 2) why 10 minutes of incubation only? Several authors recommend 24hours. Please explain this part better.
  7. Storage: Altought the results are interesting, a work with a food matrix is desirable.

Reviewer 2 Report

The manuscript aims to form complexes of α-, β- and γ-CDs with a tomato oil rich in lycopene extracted from supercritical CO2 (TO) and to comparatively evaluate the aptitude of these complexes through FTIR, DSC, antioxidant activity and encapsulation efficiency. The topic is interesting and the used methodology is applicable. In general, the manuscript is very well constructed and written. Results are well described and discussion in thorough.  Additionally, there are a few minor revision requests. Please find more detailed comments below.

Materials and Methods

Materials and methods are well described enabling the understanding of their use, purpose and even repetition of the experiments. A few minor comments:

Line 120-121. The information in topic 2.2 could be further developed and detailed.

  • I suggest changing the title of the topic to "tomato oil encapsulation process" since this topic will address how the encapsulation was carried out.
  • What was the source of the tomato? Was the tomato purchased fresh from the local market or was it already dehydrated? Was it obtained from any company or farmer?

  • The term "the dehydrated tomato matrix" refers to tomato pulp with the peel without the seeds?

  • I suggest you comment on obtaining the raw material used in the work and how it was prepared before performing the super critical extraction.

Line 143. Provide basic HPLC information. Which column was used? Which brand of HPLC? Were standard curves performed?

Line 210. Provide some explanation as to why 500mg for TO/CD emulsion and 200mg for powder

Results and discussion

The legend of all tables and figures needs to be improved.

  • The authors have to put what the letters of the statistical analysis are representing.
  • In figures that are a set of images represented by the letters a, b, c ....., the authors need to provide what each figure represents.

Table 1. Please check the statistical analysis, mainly beta-carotene related.

Reviewer 3 Report

The research results are very interesting. A few minor errors in the presentation of the results should be corrected.

Detailed comments:

  • Lines 246-248: Please add clarification: what the superscript letters mean. Are significant statistical differences in the mean values in the columns or in the rows ?
  • Lines 219-320: Please describe what is shown in figures 3a to 3i.
  • Line 336: Please describe what is shown in figures 4 from 4a to 4l.
  • Line 365: Please describe what is shown in figures 5 from 5a to 5f.

Round 2

Reviewer 1 Report

The authors have answered all the question but the antioxidant activity should be improved: Altought they said that there are no reports about ABTS at 24h, there are different articles measuring at 1, 2... and 24hs. The fact here is because the reviewer is curious about the different effect of emulsion and powder and he thinks that some antioxidant activity remains after only 6 minutes... Or maybe the different mass normalization could affect. Why didn't the authors take a sample of the dust and measure the amount of TO per gr, to normalize the sample?

I recommend to make one point more for example at 30 minutes to compare the ABTS report and to normalize the reference. It is difficult to compare now.

Author Response

We deeply thank the reviewer for further pushing us to improve our manuscript in the section discussing the antioxidant capacity of CD-encapsulated tomato oil. Thanks to his/her warm request, we revised all data relevant to our TEAC experiments and we actually caught an error in data presented in Table 1. Basically, we had previously planned to present TEAC values of emulsions and powders referring both of them to the mass of oil, to make data comparable. We calculated the mass of TO in the powder rather than measure it, due to some experimental difficulties arising from the small amounts of samples handled. A previous draft of the paper contained a table with these values. Afterwards, due to the infrared analysis results, we suspected some oil loss in the freeze-dried samples and decided to refer their TEAC values to the mass of powder. Somehow the previous table was updated only in the units, but the numerical values remained the same! We missed to catch this error and the wrong table was inserted in the submitted version although the comment in the text was relevant to the right updated TEAC values. Fortunately, the careful reviewer did not miss the inconsistency of data appearing in that table; in fact, TEAC values referred to the mass of powders are expected to be lower than emulsion values referred to the loaded TO.  We have fixed this error, inserting in the table TEAC values of powders referred to both mass of oil and total mass of powder. We have also investigated the time course of the decolorization reaction with some representative samples (Supplementary Figure 3) and we found that the TEAC at 10 min of emulsions and powders cannot be considered a total TEAC value, since residual ABTS+ neutralization capacity is present even after 3 h of incubation (a clear plateau value of the decolorization ratio was not reached). The reviewer was right also in this case. Relevant time course graphs have been inserted in the Supplementary Material. However, we avoided to perform long overnight assays to obtain total TEAC values, since the instability of the radical and of the sample itself makes poorly reliable these measurements. We decided instead to repeat all TEAC measurements with 1 h incubation time, which represents a good compromise between too fast (10 min) and overnight measurements. The table and the text have been therefore updated with these data.

Thus, the manuscript was modified as follows:

Lines 224-237: "After 1 h incubation, the tubes were centrifuged for 2 min at 7000g and the top layered organic phase was removed. Finally, the absorbance was measured at 734 nm for all samples and the antioxidant activity of TO was given as α-Tocopherol Equivalents per mass of oil. For TO/CD emulsions and TO/CD freeze-dried powders, the same procedure was applied, except that Trolox was used as hydrophilic antioxidant standard dissolved in PBS (phosphate buffered saline). All TO/CD emulsions were analyzed without dilution. After 1 h incubation step, the solutions were centrifuged 2 min at 7000g to pellet the suspended a-, b- and g-CD encapsulated TO droplets, the absorbance was measured at 734 nm for all samples and the antioxidant activity of relevant emulsions was given as Trolox Equivalents per mass of sample. For TO/a, b- and g-CD powders, samples were suspended at 10 mg/mL concentration in PBS and the TEAC was assessed following the same procedure outlined for the emulsions. Furthermore, to evaluate the time dependence of TEAC and determine possible differences among emulsions or between emulsions and powders, the antioxidant activity of some representative samples (crude TO, TO/α, β-CD powders and TO/α-CD emulsion) was monitored up to 180 min."

Lines 586-614: "A further issue concerns the kinetics of radical neutralization, which can be slowed down in a micro-heterogeneous environment affecting TEAC values in a time-dependent way. Although, the time course of the ABTS·+ decolorization reaction of all assayed representative samples (crude TO, TO/α, β-CD powders and TO/α-CD emulsion) showed a linear trend, differences in the slope of the interpolation lines were observed, indicative of either a slight dependence on CD typology or sample physical state of antioxidant kinetics (Supplementary Figure 3).

The α-Tocopherol Equivalent Antioxidant Capacity (α-TEAC) value of crude TO dissolved in hexane was 64±3 µEq/g, in rough agreement with the value of 86 µEq/g calculated on the basis of carotene and lycopene content.

The TEAC values of TO/α-CD and TO/β-CD emulsions, referred to the mass of loaded TO, were slightly higher than that of the crude TO, while a lower value was registered for TO/γ-CD (Table 2). This finding agrees with the higher efficiency of α- and β-CDs as emulsifying agents for TO compared to γ-CD. Moreover, the TEAC values of emulsions, equivalent to the α-TEAC of crude TO, indicated that the encapsulated hydrophobic antioxidants retain their ability to scavenge ABTS·+. TEAC values of suspended TO/CD powders, referred to the mass of freeze-dried sample (Table 2), were influenced by the molecular mass of specific CDs (employed at the same molar concentration in the starting emulsion) and by a possible loss of oil components during the freeze-drying process, which may affect the TO/CD mass ratios. However, the TEAC values of powders referred to the mass of loaded TO were also calculated for a more direct data comparison, considering the different CD molecular weights and neglecting any possible change of TO/CD mass ratio after emulsion freeze-drying (Table 2). These data indicate that the freeze-drying step does not result in a loss of TO antioxidant capacity, which instead appears highly stable. In the case of α-CDs and γ-CDs a slight increase of TEAC was even observed moving from the emulsion to the relevant powder. Moreover, the TEAC values of the freeze-dried powders confirmed the better performances of α- and β-CDs with respect to γ-CDs not only as oil-dispersing agents, but also as carrier systems for the vehiculation of active hydrophobic antioxidants across aqueous media. Furthermore, the measured TEAC values were in line with the antioxidant EE%, indicating that the loss of acylglycerols occurring during freeze-drying, as revealed by FTIR spectra, was not accompanied by a significant loss of oil dispersed antioxidants.".

Lines 615-621: Table 2. Antioxidant activity of TO, and TO/a-, b-and g-CD powders evaluated by TEAC assay. The antioxidant activity of TO is given as α-Tocopherol Equivalents per mass of oil; for TO/CD emulsions and powders it is expressed as Trolox Equivalents per mass of loaded oil or total powder. Different lowercase and capital letters denote significant differences among samples expressed per mass of loaded oil and total powder, respectively (Tukey post hoc test, ? < 0.05).

Samples

TEAC

µEq/g TO (µEq/g powder)

TO

64±3bc

Emulsions

TO/α-CDs

75±5ab

TO/β-CDs

73±5ab

TO/γ-CDs

54±4c

Powders

TO/α-CDs

85±9a (29±3A)

TO/β-CDs

72±6ab (22±2B)

TO/γ-CDs

68±6bc (19±2B)

Lines 723-725: "Figure S3: Time course of ABTS+ decolorization monitored at 734 nm for some representative samples. Panel A and B are referred to samples dispersed in water, while panel C and D are referred to samples dissolved in hexane. Continuous lines are linear fittings."

All changes are highlighted in red in the text.